# Effect of Plant Bioactive Compounds Supplemented in Transition Dairy Cows on the Metabolic and Inflammatory Status

**DOI:** 10.3390/molecules27186092

**Published:** 2022-09-18

**Authors:** Dana Kumprechtová, Thibaut Chabrillat, Simon Guillaume, Sylvain Kerros, Romana Kadek, Eva Indrová, Josef Illek

**Affiliations:** 1Nutrition and Feeding of Farm Animals, Institute of Animal Science, 104 00 Prague, Czech Republic; 2Phytosynthése, 632 00 Mozac, France; 3Department of Large Animal Clinical Laboratory, University of Veterinary Sciences Brno, 612 42 Brno, Czech Republic; 4Ruminant and Swine Clinic, University of Veterinary Sciences Brno, 612 42 Brno, Czech Republic

**Keywords:** plant bioactive compounds, inflammation, energy status, periparturient cows

## Abstract

(1) Background: This study evaluated the effects of a plant bioactive (Phyto Ax’Cell, Phytosynthese, Mozac, France) on the inflammatory status and health of dairy cows during calving. (2) Methods: 46 Holstein crossbred cows were randomized into a control group (CON, *n* = 23) and the Phyto Ax’Cell group (PAC, *n* = 23). PAC received Phyto Ax’Cell at 25 g/cow/day, from 15 days prepartum to 7 days postpartum. Blood analyses were performed weekly from D-7 to D14 to evaluate the energy metabolism and inflammatory status; rectal temperature was measured daily within 14 days from calving day (D0). (3) Results: PAC showed lower serum haptoglobin at D7 (0.55 vs. 0.79 mg/mL; *p* < 0.05) and D14 (0.44 vs. 0.66 mg/mL; *p* < 0.05). CON had a higher number of circulating white blood cells and granulocytes on D7 (*p* < 0.05). Fewer cows from PAC showed hyperthermia (≥39 °C) during the first 2 weeks postpartum (−7%, *p* < 0.05). Energy metabolism, which was represented by the NEFA/cholesterol ratio, improved (0.21 vs. 0.36 at D0, *p* < 0.1; 0.19 and 0.15 vs. 0.36 and 0.32, respectively, at D+7 and D+14, *p* < 0.05) under the plant bioactive supplementation. (4) Conclusions: The results suggest that the anti-inflammatory plant bioactive compound with Brazilian green propolis administered during calving had a beneficial effect on the energy and inflammatory status of dairy cows.

## 1. Introduction

High-yielding dairy cows face multiple metabolic challenges during early lactation. They are susceptible to pathologies due to the fact of a negative energy balance, inadequate supply of nutrients and immune dysregulation [1]. The period of transition from gestation to lactation is associated with numerous body adaptations to late pregnancy, colostrum synthesis, parturition processes, onset of lactation and reproductive tract recovery. Research indicates a strong link between metabolic stress induced by the transition, periparturient immune dysfunction and disease incidence [2]. Reproductive tract health is negatively influenced consequently [3]. Chebel et al. [4] studied the loss of body condition during the dry period and demonstrated its effects on health disorders and reduction of productive and reproductive performance in Holstein cows. Oxidative stress and an enhanced systemic inflammatory response may be the common factors that contribute to metabolic and infectious disorders in transition cows [5]. To elicit an effective antioxidant response and fuel immune cell activation, cows must use energy that would otherwise be directed towards milk production. Early lactation inflammation, indicated by elevated serum haptoglobin, has been associated with reduced milk yield and poor reproductive performance [6]. All of these metabolic parameters correlate to inflammation and pain in transition dairy cows and, thus, are related to the status of their well-being.

In Europe, the use of natural bioactive compounds in livestock nutrition has increased considerably over the past decades in response to legislative changes concerning the use of antibiotic growth promoters and veterinary medicines as well as consumer demand. In a recent review [7], the summarized results suggest that the use of supplemental nutraceuticals during the transition period may enhance the metabolic, immune and antioxidant system responses, allowing for the reduction of inflammation after parturition. It has been reported that natural antioxidants, such as grape polyphenols, tannins and citrus flavonoids, mitigate oxidative stress in ruminants [8]. However, only a few publications have evaluated the impact of plant bioactives on inflammatory status [9].

The objective of this study was to investigate the effects of dietary supplementation using natural bioactive compounds, selected for their antioxidant and anti-inflammatory properties, in transition dairy cows. The hypothesis was that a combination of plant bioactives administered during the transition period would reduce inflammation and oxidative stress and improve health parameters in postpartum dairy cows.

## 2. Results

### 2.1. Milk Yield and Milk Composition

No significant differences were found between the groups in the daily milk yield and milk composition (Table 1). The average daily milk yield of multiparous cows was 41.8 and 40.33 kg in the CON and PAC groups, respectively. There was a significant interaction with parity (*p* < 0.0001), and the average milk yield of multiparous cows was 35.47 kg for CON and 34.53 kg for PAC.

### 2.2. Body Condition Score

There was no difference in the body condition scores (BCS) between the two groups before and after supplementation (i.e., D-14 and D14). A trend towards a smaller decrease in the BCS was observed between D-14 and D14 (*p* = 0.07) in PAC (Table 2).

### 2.3. Energy Metabolism

The key indicators of energy metabolism are shown in Table 3. The serum level of non-esterified fatty acids (NEFAs) was significantly (*p* < 0.05) lower in the PAC group on D7 (0.46 vs. 0.65 mM and D14 (0.44 vs. 0.68 m/M) than in the CON group. On D7 and D14 postpartum, β-hydroxybutyrate (BHB) was also numerically lower in the PAC group (0.46 vs. 0.76 mM and 0.44 vs. 0.73 mM). The NEFA/cholesterol ratio was significantly lower in PAC than in CON (0.19 vs. 0.36 on D7, *p* = 0.05; 0.15. vs. 0.32 on D14, *p* < 0.05, respectively).

The increase in the levels of BHB and NEFAs between prepartum (D-7) and postpartum (D14) was greater in CON than in PAC (BHB: 0.42–1.09 mM for CON and 0.51–0.73 mM in PAC; NEFAs: 0.21–0.68 mM for CON and 0.28–0.44 mM for PAC), with a treatment × time interaction of *p* < 0.001. The NEFA/cholesterol ratio during the prepartum–postpartum shift was greater in CON than in PAC (0.11–0.32 and 0.14–0.15, respectively), with a treatment × time interaction of *p* < 0.001.

### 2.4. Inflammation Markers and Antioxidant Capacity

The PAC group showed a lower haptoglobin level at D7 (0.55 vs. 0.79 mg/mL, *p* < 0.05) and D14 (0.44 vs. 0.66 mg/mL, *p* < 0.05). No significant differences were noted in serum amyloid A (SAA), glutathione peroxidase (GSH-Px) and total antioxidant capacity (TAC) between the treatments; however, a significant (*p* < 0.001) treatment x time interaction was noted in SAA. The values are given in Table 4 and Figure 1.

### 2.5. Cell Immunity

The hematological parameters are shown in Table 5. The cows in the CON group had a tendency towards a higher circulating total WBC count on D7 and D14, and they showed significantly (*p* < 0.05) higher numbers of granulocytes on D7 (Figure 2 and Figure 3).

### 2.6. Health

*Rectal Temperature.* Fewer PAC cows showed rectal temperatures over 39 °C during the first 2 weeks (−7%, *p* = 0.029) (Figure 4).

*Uterus Position and Tonicity.* Ultrasounds on D14 revealed that 10 cows had abdominal localization of the uterus in the CON group, and 1 (uterus in the abdomen) and 2 (uterus between abdomen and pelvis) cows did in the PAC group. Three cows with a flaccid uterus (score 3) were detected in the CON group, and zero cows with a flaccid uterus were detected in the PAC group. The ultrasound examinations indicated a trend towards better uterine health, because fewer cows in PAC showed the presence of uterine content (26% vs. 48%) and more showed a better uterus position in the pelvis (80% vs. 55% at D14) (Figure 5).

## 3. Discussion

In peripartum dairy cows, an improvement in antioxidant and anti-inflammatory status would result in a better start of lactation and improved health. Hashemzadeh-Cigari et al. [10] reported that a dietary supplementation of a mixture of curcumin, carnosic acid, cinnamaldehyde and eugenol in dairy cows improved udder health as reflected in the decreased milk SCC. In our study, there was no significant interaction of treatment and parity with SCC, but a numerically lower SCC was observed for the cows that received supplementation (88.53 vs. 367.33 for multiparous and 93.90 vs. 247.17 for primiparous). It has been demonstrated that the use of salicylic-acid-derived nonsteroidal anti-inflammatory drugs (NSAIDs) may increase milk production in cows [11,12]. Because the use of NSAIDs during early lactation is considered off-label in many countries, alternatives such as phytonutrients may provide a more feasible approach. In our study, the bioactive supplementation did not influence milk production. The calving distribution (over two months) and the presence of primiparous in the groups was not the most suitable design for assessing animal performance. A significant difference was observed for parity, which can be explained by the control of tissue mobilization in primiparous which was promoted into growth as well as milk during the first lactation [13].

Copious milk production in early postpartum requires a large amount of energy, which may result in an intensive mobilization of adipose tissue. In our study, BCS loss tended to be reduced by PAC (*p* = 0.07). However, the BCS level before the start of the experiment could have been considered during the recruitment of the cows so as to limit the interaction. It has been reported that high concentrations of nonesterified fatty acids can lead to an increased state of inflammation in cows [14]. According to Drackley [15], normal NEFA levels for cows in a state of positive energy balance are lower than 0.2 mM. In both groups included in this study, the NEFA and BHB values increased, indicating a state of negative energy balance. A significant (*p* < 0.001) treatment and time interaction in NEFA and BHB levels was also noted, with the prepartum–postpartum increase being smaller in the PAC group. Trevisi et al. [2] and Huzzey et al. [6] suggested that disease cases in periparturient dairy cows were correlated with signs of accentuated inflammatory markers such as haptoglobin. Schulz et al. [16] and Kováč et al. [17] reported a positive linear correlation between haptoglobin and subclinical ketosis markers such as BHB and NEFAs. Winkler et al. [18] found that supplementation with green tea and turmeric extract from week 3 prepartum to week 9 postpartum reduced the concentration of haptoglobin (*p* < 0.10) and the concentration of cholesterol in the liver in weeks 1 and 3 postpartum (*p* < 0.05). Our study confirmed this tendency, with lower levels of NEFAs (*p* < 0.05) and haptoglobin (*p* < 0.05) in the group of cows receiving PAC. Only a weak positive linear correlation between NEFAs and haptoglobin and BHB and haptoglobin (Pearson correlation between 0.2 and 0.32) was found. The serum NEFA/cholesterol ratio was significantly lower in the PAC group, and the interaction between treatment and time was significant. Kaneene et al. [19] observed that cows with metritis had a significantly higher post-calving NEFA/cholesterol ratio. Mikulková et al. [20] found higher NEFA and BHB in cows with metritis as compared with healthy cows. One explanation could be that the cows with metritis increased their lipomobilisation rate due to the reduced feed intake, resulting in metabolic alterations in the liver and lower serum cholesterol. SYNLAB [21] stated that the NEFA/cholesterol ratio is a good indicator of liver steatosis, and they fixed 0.2 as a threshold for steatosis. In this study, the PAC group showed a lower NEFA/cholesterol ratio at D7 and D14 (less than 0.2), which might suggest improved liver metabolism, whereas the control group displayed a ratio above the steatosis threshold (NEFA/cholesterol ratio of 0.36).

A common sign of inflammation or infection is increased body temperature. In our study, there were fewer cows with rectal temperatures above 39 °C in the group treated with plant bioactive compounds. Smith and Risco [22] suggested that rectal temperatures above 102 °F (38.8 °C) are associated with postpartum disorders. Similarly, Wenz et al. [23] concluded that rectal temperatures in cows with metabolic disorders tended to be higher than 38.9 °C and lower than 39.4 °C, whereas 39.4 °C appeared to be the lower limit for infectious diseases. Benzaquen et al. [24] observed an increase in rectal temperatures from 38.8 °C up to 39.2 °C in cows with puerperal metritis. In our study, a limit of 39 °C was used to detect hyperthermia related to metabolic disorders or metritis. In the CON group, there was a higher percentage of cows that exceeded the threshold of 39.0 °C (Figure 4).

The haptoglobin reduction was in line with the numbers of cows exhibiting signs of clinical metritis and the presence of uterine content (26% vs. 48% in the PAC and CON groups, respectively). Some studies have confirmed the association between the serum concentration of haptoglobin and the uterine infection postpartum [25,26]. Huzzey [27] reported that cows with mild or severe metritis had higher haptoglobin concentrations than healthy cows between D0 and D+12 postpartum and suggested that increased circulating haptoglobin may be used as an early indicator of metritis.

In addition, total circulating leucocytes showed a trend (*p* < 0.1) towards being lower in the PAC group from D-7 to D+14, as well as granulocyte counts and percentages, which may also indicate a lower inflammatory status in the PAC group, especially at the uterine level. Kim et al. [28] confirmed that cows that developed endometritis had significantly higher total leukocyte, neutrophil, lymphocyte and monocyte counts than control cows.

These effects on inflammation and immune modulation can be attributed to several components of PAC. In dairy cows, green propolis has previously demonstrated effects on the rumen microbiota [29] and antioxidant capacity of milk [30]. Several studies may explain its anti-inflammatory activity. Hori et al. [31] demonstrated that a Brazilian green propolis extract (EPP-AF) reduced the secretion of IL-1β by inhibiting the NLRP3 inflammasome activation in mouse macrophages. In parallel, Kelly et al. [32] reported that the inflammatory response in the endometrium of postpartum dairy cows is characterized by NLRP3 inflammasome activation in the epithelial cells, with the release of pro-inflammatory cytokine, IL-1β, leading to the recruitment of immune cells. Other authors concluded that a better understanding of the molecular regulatory mechanisms underpinning inflammasome activation within the endometrium could lead to an effective immunotherapeutic target for the prevention of uterine diseases in cattle. Similarly, artepillin C-rich propolis acts on the immune response with a significant change in the immune cells ratio (i.e., lymphocytes/neutrophils/macrophages), explained by a change in cytokines production such as IL-6 and TNF-α [33]. More recently, Nishikawa et al. [34] suggested that artepillin C with curcumins may exert effects on the activation of macrophages as a more reactive immune system. A complementary activity of artepillin C-rich propolis with curcuminoids from turmeric roots, as explained by Wang et al. [35], indicated a significant effect of curcuminoids on the immune response to lipopolysaccharide (LPS) challenge quantified via IL-1β and IL-10 measurement.

Additionally, the PAC supplementation provided plant antioxidants such as carnosic acids and polyphenols. Phenolic compounds have been reported to increase endogenous antioxidants including superoxide dismutase, catalase and GSH-Px [36]. Among polyphenols, rosemary has been reported to improve antioxidant status and reduce lipid peroxidation in sheep [37] and mid-lactation dairy cows [38]. The antioxidant status indicators measured in our study (i.e., total antioxidant capacity and glutathione peroxidase) were not significantly improved by the plant bioactive supplementation (TAC: *p* = 0.08 at D+7 and D+14; GSH-Px: *p* > 0.1). Collection of blood samples before the start of the experiment would have been useful in supporting the absence of statistical differences among groups prior to the experiment and to establish a more precise and accurate analysis of the bioactive supplementation effect.

## 4. Materials and Methods

### 4.1. Animals, Treatments and Diets

The study was conducted on a dairy farm with 600 Holstein crossbred cows (Holstein x Czech Siemental), located in the Czech Republic. Forty-six cows were used as experimental animals in this study: 22 primiparous and 24 multiparous, evenly distributed between the groups. The cows were divided into two experimental treatments: a control group (CON, *n* = 23) and a group supplemented with Phyto Ax’Cell (PAC, *n* = 23), allocated according to their day of expected calving and parity (1.69 and 1.61, respectively) (Table 6). The PAC group received PAC at 25 g/cow/day, according company recommendations and previous studies using some PhytoAx’Cell components [38], top dressed on the total mixed ration, from approximately 14 days pre-calving to 7 days post-calving. The CON cows received the basal TMR with no top dressing.

From D-21 prepartum to parturition, the cows were housed loose in a bedded pack barn with 2 closed-up pens with deep straw bedding and 6 animals per pen. The PAC and CON cows were housed separately in 2 adjacent pens of identical size and number of animals. Primiparous and multiparous animals were housed together. After calving, the cows were transferred into a free-stall pen with cubicles bedded with dry manure solids and chopped straw, treated with limestone, with approximately 10 cows per pen, divided into 2 identical compartments (PAC and CON).

The animals received a close-up diet (from D-21 to D0) and a fresh cow diet (from D0 to D60), provided twice a day (7 a.m. and 4 p.m.) as a total mixed ration (Table 7). Feed was pushed up every 2–3 h. The amount of feed offered was adjusted to allow for approximately five percent feed refusal. Feed intake was not measured.

Health was monitored daily by visual inspection, rectal temperature measurements, (D0-D14) and assessment of the data from the milking parlor.

Animal handling followed the Directive 2010/63/EU on the protection of animals used for scientific purposes and Act number 246/1992 Coll. of Laws of the Czech Republic on the protection of animals against cruelty as amended.

### 4.2. Supplement Composition

PAC was supplied by Phytosynthese, France. It is a feed supplement based on standardized plants, plant extracts and green propolis extract. The active compounds contained in the product were analyzed at the Phytosynthese LAB (Mozac, France) and Apis Flora (Ribeiro Preto, Brazil) for artepillin C and AGROBIO (Bruz, France) for total polyphenols. The product composition is provided in Table 8, in addition to crude constituents provided by the supplier: 5.0% protein, 23.4% crude fiber, 4.4% fat, 4.8% ash.

*Curcuminoids.* Quantification of the curcuminoid content of PAC was performed using HPLC (Shimadzu, Prominence series, JPN). A 1 g sample of PAC was mixed with 20 mL of methanol and boiled under reflux in a condenser for 2 h. After cooling, the mixture was filtered through cotton. The residue in the filter was recuperated, and the 2 h extraction procedure was repeated. The 2 supernatants were combined in a 50 mL volumetric flask. An aliquot of the solution was filtered through a 0.45 µm nylon filter and injected into a C-18 column (dimensions: 250 × 4.6 mm; particle size: 5 μm). The solvents for elution were acetic acid/water, acetonitrile and methanol. The amount of curcuminoids was calculated by external calibration of the reference standard of curcumin, and the result was expressed as curcumin.

*Carnosic Derivatives.* Quantification of the carnosic derivatives content of PAC was performed using HPLC (Shimadzu, Prominence series, JPN). A 20 g sample of PAC was extracted with 200 mL of acetone for 5 h in a Soxhlet extractor. After cooling, the solvent was evaporated to dryness without exceeding 60 °C. The total dry extract quantity was evaluated, and 0.1 g was diluted in 10 mL of methanol. An aliquot of the solution was filtered through a 0.45 µm nylon filter and injected onto a C-18 column (dimensions: 250 × 4.6 mm; particle size: 5 μm). The solvents for elution were acetic acid/water, acetonitrile and methanol. The amount of carnosic derivatives was calculated by external calibration of the reference standard of carvacrol and expressed as carvacrol.

*Naringin Flavonoid.* Quantification of the naringin content of PAC was performed using HPLC (Shimadzu, Prominence series, JPN). A 2 g sample of PAC was mixed with 40 mL of methanol and sonicated for 15 min in an ultrasonic bath. The mixture was filtered through cotton in a 50 mL volumetric flask completed with methanol. An aliquot of the solution was filtered through a 0.4 µm nylon filter and injected onto a C-18 column (dimensions: 250 × 4.6 mm; particle size: 5 μm). The solvents for elution were acetic acid/water and acetonitrile. The amount of naringin flavonoid was calculated by external calibration of the reference standard of naringin and expressed as naringin.

*Salicylic Derivatives*. Quantification of the salicylic derivatives content of PAC was performed using HPLC (Agilent Technologies, 1260 series, Santa Clara, CA, USA). A 3 g sample of PAC was used and analyzed according to the European Pharmacopoeia Monograph of willow bark (01/2013:1583). The amount of the salicylic derivatives was calculated by external calibration of the reference standard of salicin and expressed as salicin.

*Artepillin C.* The green propolis extract used in the mixture was analyzed by HPLC using a Shimadzu apparatus equipped with a CBM-20A controller, a LC-20AT quaternary pump, an SPD-M 20A diode-array detector and Shimadzu LC solution software, version 1.21 SP1. A Shimadzu Shim-Pack CLC-ODS column (4.6 × 250 mm; particle diameter of 5 μm; pore diameter of 100 Å) was used. The mobile phase consisted of methanol (B) and a solution of water–formic acid (0.1%, v/v) at pH 2.7 (A). The method consisted of a linear gradient of 20–95% of B over a period of 77 min at a flow rate of 0.8 mL/min. Detection was set at 275 nm.

Propolis extract was diluted with 5 mL of methanol (HPLC grade) in 10 mL volumetric flasks, subjected to sonication for 10 min and filled to volume with Milli-Q water. The samples were filtered through a 0.45 µm filter before analysis.

*Total Polyphenols.* Quantification of the total polyphenol content of PAC was performed using a spectrophotometer at a 280 nm wavelength. The number of total polyphenols was expressed as catechins.

### 4.3. Measurements and Sample Collection

*Milk Yield and Composition.* Cows were milked twice a day in a fishbone parlor for 2 × 13 cows (Fullwood Ltd., UK). Daily milk yield was recorded electronically via the herd management system, Crystal (Fullwood Ltd., UK). Milk samples were collected monthly, using Fullwood sampling units as aliquots from one evening and subsequent morning milking and pooled for further analysis. Bronopol (2-bromo-2-nitropropane-1,3-diol) preservative-treated milk samples were transported to the regional laboratory of the milk recording organization (CMSCH a.s.—LRM Brno, CZ). Concentrations of fat, lactose, protein and urea were analyzed by infrared spectrophotometry and the somatic cell count by flow cytometry using Combi Foss (Foss Electric, Hillerød, Denmark). Energy-corrected milk was calculated using the equation of Sjaunja et al. [39]:ECM = milk (kg) × [0.383 × fat (%) + 0.242 × protein (%) + 0.7832]/3.1138

*Body Condition Score.* BCS was estimated for each cow included in the study approximately at days −14, 14 and 40 post-calving. The BCS was evaluated on a scale from 1 (emaciated) to 5 (overweight) in increments of 0.25 according to Edmonson et al. [40].

*Blood Sampling.* Blood samples were collected at weekly intervals from 1 week prepartum to 2 weeks postpartum, from the vena coccygea mediana, using HEMOS tubes (Hemos H-02, Gama Group, Czech Republic) for serum, HEMOS tubes containing heparin for whole blood (GSH-Px activity measurement) and tubes containing EDTA as an anticoagulant for whole blood samples [41]. Blood samples for glucose measurement were taken into the tubes with sodium fluoride and taken to the laboratory where they were analyzed. Centrifuged serum was frozen at −20 °C until processing. Hematology analyses were performed on whole blood within 2 h of collection. The blood samples were always taken between 9:00 and 10:00 a.m.

### 4.4. Laboratory Analyses

Laboratory analyses were performed at the Central Laboratory of the University of Veterinary and Pharmaceutical Sciences Brno.

Whole blood or serum samples were analyzed for the following:D-7 pre-calving: NEFAs, BHB, glucose, cholesterol, total protein, albumin, urea and calcium;D0 calving day: NEFAs, BHB, glucose, cholesterol, total protein (TP), albumin, urea, TAC, GSH-Px, calcium, haptoglobin and SAA; hematology;D7 post-calving: NEFAs, BHB, glucose, cholesterol, TP, albumin, urea, TAC, GSH-Px, calcium, haptoglobin and serum amyloid A; hematology;D14 post-calving: NEFAs, BHB, glucose, cholesterol, TP, albumin, urea, TAC, GSH-Px, calcium, haptoglobin and serum amyloid A; hematology.

Urea, glucose, total protein and total cholesterol were measured by photometric methods using an automatic analyzer Cobas Mira (Roche Diagnostics, Basel, Switzerland). The following analytical sets were used: urea UV KIN 4 × 50, Cat. No. 1307017 (Erba Lachema s.r.o., CZ); lGlukosa, Cat. No. 11601; LProtein tital, Cat. No. 612751; LCholesterol, Cat. No. 10851; LAST, Cat. No. 10351 (BioVendor a.s., CZ). Serum NEFAs and BHB were measured with standardized kits (Randox Laboratories Ltd., Crumlin, UK; NEFAs, Cat. No. FA 115; BHB: RANBUT, Cat. No., RB 1008). Serum haptoglobin was analyzed by colorimetry (Konelab 20XT), SAA by Sandwich ELISA (Agilent Technologies, formerly BioTek Instruments, Inc., Santa Clara, USA). GSH-Px activity was measured in whole heparinized blood with the RANSEL kit (Randox Laboratories Ltd., Crumlin, UK) using a UV method based on that of Paglia and Valentine [42]. The method consisted of measuring the decrease in the absorbance at 340 nm due to the fact of NADPH oxidation by the reaction with glutathione reductase (GR). For the determination of GSH-Px, the automatic biochemical analyzer Konelab 20XT (Thermo Fisher Scientific, Vantaa, Finland) was used.

Hematology was performed with the analyzer BC-2800 Vet (Mindray Bio-Medical Electronics, Guangdong, China).

### 4.5. Clinical Parameters

*Rectal Temperature.* Rectal temperature was measured daily (8 a.m.) from D0 (calving) to D14 in all the cows under study.

*Vaginal Discharge.* From D0 (calving) to D14, daily (8 a.m.) observations were performed. Cows showing a watery, purulent or brown and fetid vaginal discharge and rectal temperature ≥39.0 °C were diagnosed as having clinical metritis [20].

Uterine health and involution. The uterus was examined at D13 and D25 by ultrasound (5 MHz transrectal linear transducer, Aloka SSD-500, Aloka Co., Ltd., Japan) for the presence of uterine content and position of the uterus (abdomen/pelvis).

Per rectum palpation was conducted to establish a uterine score (3 = a flaccid uterus larger than one hand; 2 = a uterus with moderate tonicity and smaller than one hand; 1 = a high tonicity and fewer than three fingers in width [43]).

### 4.6. Statistical Analyses

Data were analyzed with the XLSTAT software (Addinsoft, 19.4 version). The cow was considered as the experimental unit, and treatment was used as a fixed effect. The normality of the distribution was analyzed with the Shapiro–Wilk test, and homoscedasticity was checked with Fisher’s test. According to these results, the parametric comparison was evaluated with a Student’s *t*-test, and nonparametric comparisons were conducted with the Mann–Whitney test; *p* < 0.05 was interpreted as significantly different. Hyperthermia was analyzed with Codran’s Q test.

## 5. Conclusions

Recently, there has been growing research and interest in phytonutrients in dairy cows. The objective of this study was to evaluate the effects of plant secondary metabolites with reported anti-inflammatory properties on transition dairy cows. The results show the plant bioactive complex could support metabolism of dairy cows during the periparturient period. We observed a reduction in serum haptoglobin postpartum and a better stability of serum parameters over time, indicating a mitigation of the inflammation. Regarding energy status, we observed a reduction in serum NEFAs post-partum suggesting a lower mobilization of adipose tissue. Plant bioactive compounds with antioxidant and anti-inflammatory properties administered during the transition period seem to improve health parameters in postpartum dairy cows.

Further research is needed to fully understand the effects of plant bioactive compounds on the transition cow and their modes of action.

## Figures and Tables

**Figure 1 molecules-27-06092-f001:**
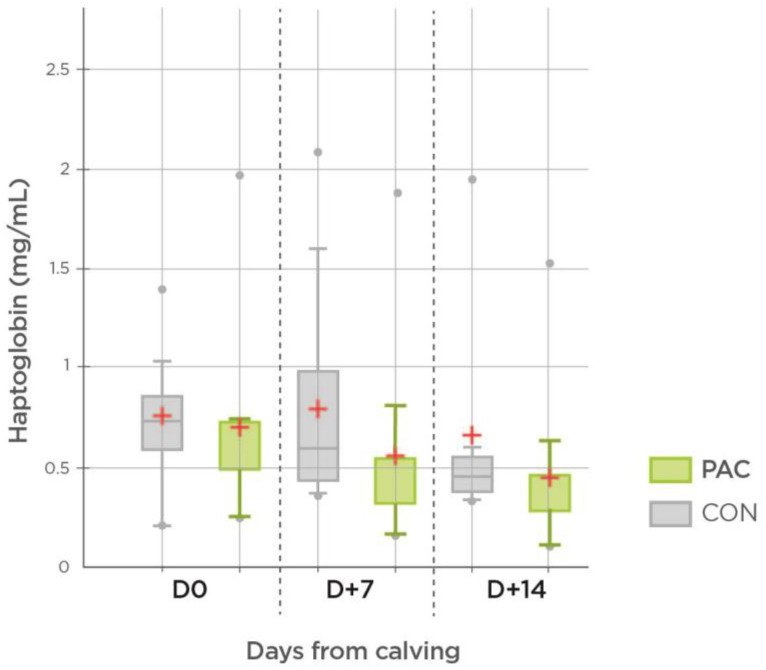
Effect of pant bioactive compounds on serum haptoglobin levels (*n* = 23).

**Figure 2 molecules-27-06092-f002:**
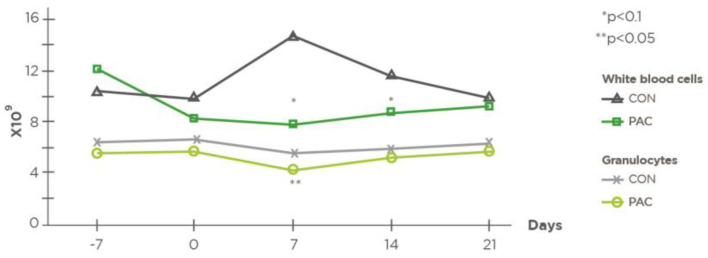
Effect of plant bioactive compounds on total number of white blood cells and granulocytes (*n* = 23).

**Figure 3 molecules-27-06092-f003:**
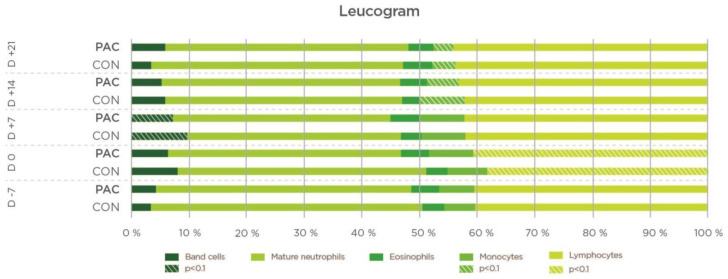
Effect of plant bioactive compounds on leucogram (*n* = 23).

**Figure 4 molecules-27-06092-f004:**
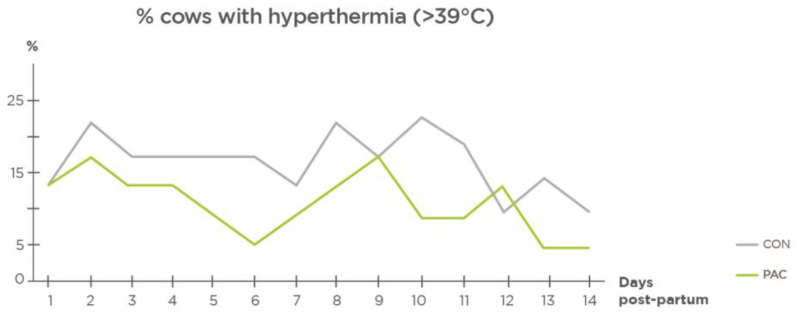
Effect of plant bioactive compounds on cows with hyperthermia within 0–14 days postpartum (%) (*n* = 23).

**Figure 5 molecules-27-06092-f005:**
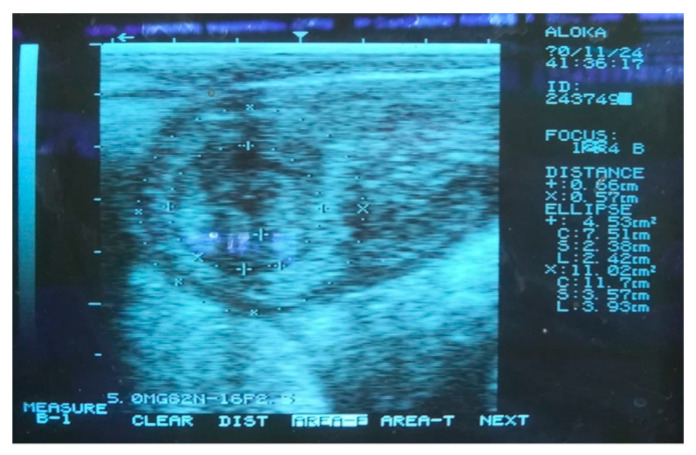
Cross-sectional ultrasound image of the uterine horn (Aloka SSD-500).

**Table 1 molecules-27-06092-t001:** Average milk yield and composition from the first three individual milk tests.

	Treatment							
	CON ^1^ (*n* = 23)		PAC ^2^ (*n* = 23)			*p*-Value		
**Item**	**Multiparous**	**Primiparous**	**Multiparous**	**Primiparous**	**SEM**	**Treatment**	**× Parity**	**Treatment × Parity**
Milk yield (kg/d)	41.8	35.47	40.33	34.53	6.57	0.345	<0.0001	<0.0001
Fat (%)	4.02	3.92	3.93	3.86	0.24	0.936	0.316	0.145
Protein (%)	3.19	3.36	3.29	3.23	0.44	0.577	0.291	0.665
Lactose (%)	5.01	5.13	5.01	5.13	0.17	0.884	0.001	0.010
ECM ^3^ (kg/d)	41.41	35.00	40.75	33.33	6.88	0.421	<0.0001	0.000
Somatic Cell Count	367.33	247.17	88.53	93.90	248.62	0.157	0.697	0.470

^1^ CON = control diet; ^2^ PAC = control diet + 25 g/cow/day of Phyto Ax’Cell; ^3^ ECM = energy-corrected milk.

**Table 2 molecules-27-06092-t002:** Body condition scores of cows receiving plant bioactive compounds from D-15 to D+7 relative to parturition.

	Body Condition Score (BCS)					BCS Evolution
	D-14	D14	D40	*p* (Time)	*p* (Treatment × Time)	D14 minus D-14
CON ^1^	3.55 ^a^	2.97 ^b^	2.87 ^b^	<0.01		−0.58
PAC ^2^	3.49 ^a^	3.07 ^b^	2.89 ^b^	<0.01		−0.42
SEM	0.13	0.19	0.18			0.21
*p* (Treatment)	0.327	0.256	0.739		<0.01	0.073

^1^ CON = control diet; ^2^ PAC = control diet + 25 g/cow/day of Phyto Ax’Cell. ^a,b^ Differed significantly over time, *p* < 0.05.

**Table 3 molecules-27-06092-t003:** Serum metabolite concentrations in cows receiving plant bioactive compounds.

Parameter		Day -7	Day 0	Day 7	Day 14	*p* (Time)	*p* (Treatment × Time)
**Total protein**	CON ^1^	73.24 ^ab^	69.54 ^a^	73.44 ^ab^	77.80 ^b^	0.002	
**(g/L)**	PAC ^2^	72.28	71.82	74.17	77.56	0.10	
	SEM	5.76	5.96	5.45	6.40		
	*p* (Treatment)	0.500	0.370	0.670	0.890		0.004
**Albumin**	CON	36.33	35.39	35.81	37.00	0.30	
**(g/L)**	PAC	36.03	35.44	35.98	36.55	0.16	
	SEM	1.85	2.07	2.07	2.04		
	*p* (Treatment)	0.930	0.950	0.410	0.590		0.23
**Globulins**	CON	36.91 ^ab^	34.15 ^a^	37.63 ^ab^	40.80 ^b^	0.04	
**(g/L)**	PAC	36.26	36.38	39.19	41.01	0.16	
	SEM	6.21	5.89	6.34	7.16		
	*p* (Treatment)	0.772	0.379	0.501	0.930		0.06
**Albumin/globulin ratio**	CON	1.030	1.076	0.994	0.947	0.34	
	PAC	1.036	1.028	0.927	0.929	0.19	
	SEM	0.199	0.196	0.184	0.187		
	*p* (Treatment)	0.770	0.370	0.310	0.950		0.27
**Glucose**	*p* (Treatment)	3.51 ^a^	3.43 ^ab^	3.17 ^b^	3.10 ^b^	0.05	
**(mM)**	PAC	3.69 ^b^	3.93	3.26 ^b^	3.20 ^b^	<0.001	
	SEM	0.21	0.53	0.42	0.34		
	*p* (Treatment)	0.020	0.1500	0.54	0.870		0.001
**Blood urea**	CON	4.79 ^c^	4.49 ^bc^	3.69 ^a^	3.81 ^ab^	0.001	
**(mM)**	PAC	4.71 ^b^	4.13 ^a^	3.89 ^a^	4.04 ^a^	0.01	
	SEM	0.82	0.21	0.50	0.51		
	*p* (Treatment)	0.810	0.130	0.310	0.240		<0.001
**Cholesterol**	CON	1.982 ^a^	1.72 ^a^	2.03 ^a^	2.69 ^b^	<0.001	
**(mM)**	PAC	2.30 ^ab^	2.00 ^a^	2.50 ^bc^	2.99 ^c^	<0.001	
	SEM	0.31	0.42	0.43	0.55		
	*p* (Treatment)	0.012	0.460	0.003	0.140		<0.001
**BHB ^3^**	CON	0.42 ^a^	0.69 ^b^	1.11 ^b^	1.09 ^b^	<0.001	
**(mM)**	PAC	0.51 ^a^	0.56 ^ab^	0.76 ^c^	0.73 ^bc^	<0.001	
	SEM	0.31	0.19	0.46	0.44		
	*p* (Treatment)	0.150	0.470	0.870	0.530		<0.001
**NEFAs ^4^**	CON	0.21 ^a^	0.57 ^b^	0.65 ^b^	0.68 ^b^	<0.001	
**(mM)**	PAC	0.28	0.34	0.46	0.44 ^a^	0.12	
	SEM	0.16	0.26	0.23	0.31		
	*p* (Treatment)	0.920	0.230	0.045	0.030		<0.001
**NEFA/cholesterol** **ratio**	CON	0.11 ^a^	0.36 ^b^	0.36 ^b^	0.32 ^b^	<0.001	
	PAC	0.14 ^a^	0.21 ^b^	0.19 ^b^	0.15 ^b^	0.01	
	SEM	0.09	0.18	0.15	0.17		
	*p* (Treatment)	0.422	0.084	0.005	0.034		<0.001

^1^ CON = control diet; ^2^ PAC = control diet + 25 g/cow/day of Phyto Ax’Cell; ^3^ BHB = β-hydroxybutyrate; ^4^ NEFAs = non-esterified fatty acids; ^a, b, c^ Differed significantly over time, *p* < 0.05.

**Table 4 molecules-27-06092-t004:** Inflammation and antioxidant capacity markers in cows receiving plant bioactive compounds.

Parameter		Day 0	Day 7	Day 14	*p* (Time)	*p* (Treatment × Time)
**TAC** ^3^	CON ^1^	0.78 ^a^	0.81 ^a^	0.85 ^b^	0.03	
**(mM)**	PAC ^2^	0.83	0.85	0.88	0.25	
	SEM	0.11	0.08	0.08		
	*p* (Treatment)	0.830	0.120	0.640		0.09
**SAA** ^4^	CON	55.00 ^a^	38.22 ^ab^	27.19 ^b^	0.01	
**(mg/mL)**	PAC	43.97 ^a^	28.58 ^ab^	20.63 ^b^	0.01	
	SEM	21.21	20.72	17.92		
	*p* (Treatment)	0.130	0.120	0.170		0.00
**Haptoglobin**	CON	0.76 ^b^	0.79 ^b^	0.66 ^a^	0.05	
**(mg/mL)**	PAC	0.70 ^b^	0.55 ^ab^	0.44 ^a^	0.01	
	SEM	0.26	0.35	0.28		
	*p* (Treatment)	0.170	0.035	0.027		0.06
**GSH-Px** ^5^	CON	1095.74	1077.47	1061.32	0.74	
**(ukat/L)**	PAC	1065.15	1028.31	1000.51	0.20	
	SEM	99.24	93.02	101.23		
	*p* (Treatment)	0.470	0.160	0.110		0.13

^a,b^ Differed significantly over time, *p* < 0.05. ^1^ CON = control diet; ^2^ PAC = control diet + 25 g/cow/day of Phyto Ax’Cell. ^3^ TAC = total antioxidant capacity; ^4^ SAA = serum amyloid A; ^5^ GSH-Px = glutathione peroxidase.

**Table 5 molecules-27-06092-t005:** Hematological parameters in cows receiving plant bioactive compounds.

Parameter		Day-7	Day 0	Day 7	Day 14	*p* (Time)	*p* (Treatment × Time)
**Leucocytes** **(× 10^9^/L)**	CON ^1^	10.62	9.82	14.81	11.54	0.284	
	PAC ^2^	12.14	8.34	7.84	8.75	0.189	
	SEM	3.74	2.26	6.42	2.77		
	*p* (Treatment)	0.08	0.13	0.08	0.08		0.583
**Lymphocytes** **(× 10^9^/L)**	CON	3.29	2.74	3.54	3.26	0.239	
	PAC	2.77	2.63	2.80	2.69	0.902	
	SEM	0.96	0.53	1.28	0.90		
	*p* (Treatment)	0.06	0.83	0.78	0.63		0.389
**Monocytes** (**× 10^9^/L)**	CON	0.95	0.88	0.81	0.90	0.201	
	PAC	0.86	0.84	0.77	0.89	0.363	
	SEM	0.20	0.23	0.2	0.24		
	*p* (Treatment)	0.21	0.87	0.52	0.98		0.333
**Granulocytes** **(× 10^9^/L)**	CON	6.38	6.63	5.66	5.90	0.695	
	PAC	5.58	5.67	4.27	5.17	0.066	
	SEM	1.36	1.70	1.85	1.34		
	*p* (Treatment)	0.130	0.180	0.035	0.150		0.007
**Lymphocytes (%)**	CON	29.00	27.95	30.08	38.90	0.789	
	PAC	30.11 ^a^	30.20 ^a^	38.11 ^b^	31.51 ^a^	0.039	
	SEM	7.08	7.51	9.79	12.37		
	*p* (Treatment)	0.970	0.320	0.020	0.210		0.513
**Monocytes (%)**	CON	9.05	8.82	9.28	9.68	0.068	
	PAC	9.53	9.28	10.09	10.62	0.096	
	SEM	1.35	1.45	1.44	1.32		
	*p* (Treatment)	0.300	0.120	0.300	0.066		0.042
**Granulocytes (%)**	CON	60.26	63.2	60.6	59.87	0.580	
	PAC	61.60 ^b^	60.5 ^b^	51.8 ^a^	58.10 ^b^	0.017	
	SEM	5.96	7.93	10.09	6.94		
	*p* (Treatment)	0.620	0.400	0.017	0.058		0.040
**Hemoglobin** **g/L**	CON	103.09 ^c^	99.7 ^b^	95.8 ^ab^	94.70 ^a^	0.006	
	PAC	100.17 ^c^	95.3 ^c^	93.5 ^b^	89.83 ^a^	<0.001	
	SEM	7.29	8.37	6.60	6.68		
	*p* (Treatment)	0.410	0.860	0.370	0.043		0.004
**Erythrocytes** **(× 10^12^/L)**	CON	6.92	6.55	6.42	6.32	0.053	
	PAC	6.80	6.73	6.57	6.40	0.078	
	SEM	0.61	0.66	0.55	0.56		
	*p* (Treatment)	0.880	0.270	0.200	0.660		0.330
**Hematocrit** **(%)**	CON	34.61	35.10	33.80	34.51	0.498	
	PAC	34.42 ^a^	34.20 ^a^	33.01 ^b^	31.37 ^b^	0.002	
	SEM	2.82	3.01	2.58	4.79		
	*p* (Treatment)	0.877	0.498	0.440	0.059		0.155
**Platelets** **(× 10^9^/L)**	CON	255.91 ^a^	295.4 ^a^	338.6 ^b^	561.48 ^c^	<0.0001	
	PAC	271.09 ^a^	310.2 ^ab^	339.4 ^bc^	340.30 ^c^	0.045	
	SEM	88.00	72.69	58.11	182.26		
	*p* (Treatment)	0.670	0.655	0.974	0.217		0.047

^1^ CON = control diet; ^2^ PAC = control diet + 25 g/cow/day of Phyto Ax’Cell. ^a, b, c^ Differed significantly over time, *p* < 0.05.

**Table 6 molecules-27-06092-t006:** Randomization of the experimental cows (mean values and standard errors of the mean).

Item	CON^1^ (*n* = 23)	PAC ^2^ (*n* = 23)	SEM	*p*
Average parity rank	1.69	1.61	0.11	NS ^3^
Number of heifers	11	11	-	-
Number of cows	12	12	-	-

^1^ CON = Control Diet; ^2^ PAC = Control Diet + 25 g/cow/day of Phyto Ax´Cell; ^3^ NS = statistically non-significant.

**Table 7 molecules-27-06092-t007:** Composition of the control diet on a fresh matter basis ^1^.

Ingredients	Close-Up Diet (kg)	Fresh Cow Diet (kg)
Maize silage	19.0	22.0
Alfalfa silage	2.0	3.0
Whole crop rye silage	0.0	4.0
Barley straw	2.0	0.0
Alfalfa hay	0.0	1.0
Brewers’ grains	1.5	6.0
High moisture corn	0.0	2.0
Rapeseed meal	0.6	0.0
WDGS ^3^	0.0	1.0
Concentrate ^2^	2.8	5.8
Palmitate	0.0	0.15
MP ion (anionic salts)	0.5	0.0
Water	3	4
Total fresh matter/cow/day	31.40	48.95

^1^ Analytical composition of the total mixed ration: 43.87%, dry matter, on a dry matter basis; 15.35%, crude protein; 23.41%, starch; 6.18%, ash; 34.61%, NDF; 19.58%, ADF. ^2^ Concentrate composition on a DM basis: 70.90%, rapeseed meal; 19.35%, ground barley; 2.4%, potassium carbonate; 1.6%, salt; 2.4%, limestone; 1.0%, caustic magnesite; 0.75% mineral mix (Zemos VVS, Vermerovice, Czech Republic; Analytical values (per kg): calcium, 8.47%; sodium, 2.0%; magnesium, 4.57%; phosphorus, 0.49%;). Additives: calcium iodate (Ca(IO_3_)_2_), 15.0 mg; vitamin A, 165,000 IU; vitamin D3, 50,000 IU; vitamin E, 2300 mg (alpha-tocopherol 2010 mg); copper sulphate pentahydrate (CuSO_4_·5H_2_O), 500 mg; manganese oxide (MnO), 1000 mg; zinc oxide (ZnO), 800 mg; cobalt acetate tetrahydrate (Co(CH_3_COO·4H_2_O)), 5 mg; sodium selenite (Na_2_SeO_3_), 15 mg. Technological additives: BHT, 90 mg; BHA, 30 mg. ^3^ Wet distillers’ grains with solubles.

**Table 8 molecules-27-06092-t008:** Composition of Phyto Ax’Cell.

Compound	Content (mg/kg)
Curcuminoids	3290
Carnosic derivatives	5010
Salicylic derivatives	6360
Flavonoids (naringin)	9665
Artepillin C	180 ^1^
Total polyphenols	20,900

^1^ Calculated from ingredient analysis and the ingredient incorporation rate.

## Data Availability

The data presented in this study are available upon request from the corresponding author. The data are not publicly available due to the fact of privacy reasons.

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
