# Peer review of "Effect of Plant Bioactive Compounds Supplemented in Transition Dairy Cows on the Metabolic and Inflammatory Status"

_molecules, 2022, doi:10.3390/molecules27186092_

Round 1
Reviewer 1 Report
The present study was quite interesting research work with natural feed additive source in ruminant nutrition on their health performance.
Line no.: bioactive compounds
Avoid multiple citations for one sentence. If it’s necessary, describe each references individual.
Clearly mention the number of animals [study design] in the groups and experimental period. Check abstract and methodology sections.
In table 1 – mention the number of animals (n = ?)
Abbreviations must be defined at first mention and used consistently thereafter in entire manuscript. Eg. Line no. 166
Line no.: 134-135 and 399-400. Remove the repeated sentences
Suggested for proofread correction to rectify few grammatical errors.
Author Response
Response to Reviewer 1 Comments
Dear Reviewer,
Thank you very much for reviewing our manuscript Effect of plant bioactive compounds supplemented in transition dairy cows on the metabolic and inflammatory status. Please, find below our responses to your comments and suggestions. The manuscript has been edited accordingly.
Sincerely,
Dana Kumprechtová, on behalf of the team of authors
Point 1: Avoid multiple citations for one sentence. If it’s necessary, describe each references individual.
Response 1: We´ve tried to keep the Introduction chapter brief, in our opinion there´s no necessity to describe each reference in a separate sentence. Should we remove some references or prolong the Introduction?
Point 2: Clearly mention the number of animals [study design] in the groups and experimental period. Check abstract and methodology sections.
Response 2: Numbers of animals in the groups and experimental period are now mentioned in Abstract (L14, 15, 16), in Materials and methods (number of cows added in L244; experimental period in L248).
Point 3: In table 1 – mention the number of animals (n = ?)
Response 3: Number of animals (CON: n=23, PAC: n=23) has been added in Table 1.
Point 4: Abbreviations must be defined at first mention and used consistently thereafter in entire manuscript. Eg. Line no. 166
Response 4: The manuscript has been edited accordingly.
Point 5: Line no.: 134-135 and 399-400. Remove the repeated sentences
Response 5: The manuscript has been edited accordingly.
Point 6: Suggested for proofread correction to rectify few grammatical errors.
Response 6: The manuscript has been edited by a professional MDPI editor and native English speaker. The English Editing Certificate is provided.

Reviewer 2 Report
rows 171-172 (page 7) specify if the correletion is on coplete database and report in brackets the value (r= ?!?)
rows 191-192 (page 7) add at the end of the sentence: (Figure 4)
row 239 what authors mean with crossbreed? if what I think, please indicate the second breed
the work it is suitable for publication as it. i only suggest the authors for next experiments to collect blood samples from jugular vein and before meal, it will be more easy to compare with result in literature
Author Response
Response to Reviewer 2 Comments
Dear Reviewer,
Thank you very much for reviewing our manuscript Effect of plant bioactive compounds supplemented in transition dairy cows on the metabolic and inflammatory status. Please, find below our responses to your comments and suggestions. The manuscript has been edited accordingly.
Sincerely,
Dana Kumprechtová, on behalf of the team of authors
Point 1: rows 171-172 (page 7) specify if the correlation is on complete database and report in brackets the value (r= ?!?)
Response 1: Yes, the correlation was on complete database and ranged between 0.2-0.32 – added in the text, L173.
Point 2: rows 191-192 (page 7) add at the end of the sentence: (Figure 4)
Response 2: Done (currently L193).
Point 3: row 239 what authors mean with crossbreed? if what I think, please indicate the second breed.
Response 3: Holstein x Czech Siimmental cross (added L240, 241)
Point 4: I only suggest the authors for next experiments to collect blood samples from jugular vein and before meal, it will be more easy to compare with result in literature.
Response 4: Thank you. We´ll take this into account our future trials.

Reviewer 3 Report
General comments
l This study aims to identify a commercial plant bioactive (Phyto Ax’Cell, Phy-12 tosynthese, France) on the inflammatory status and health of dairy cows around calving. This research is helpful for dairy cow production. However, the results need to be revised before acceptance. More information should be added to the plant bioactive (Phyto Ax’Cell, Phy-12 tosynthese, France).
The limitations.
1) Please list out the detailed animal information such as DIM, milk yield, and parity in average +-SD. You have 22 primiparous and 24 multiparous cows, are they allocated to the two groups on average?
2) Was the feed intake measured?
3) Why do you select 25 g/cow/day? Did you conduct a pre-trial or refer to a previous study to use this dose?
4) Suggest changing Table 6 to percentage of dry matter basis, not the kg in the fresh matter.
5) The total polyphenols in Phyto Ax’Cell is 20.9g/kg, so what are the other components?
6) The first sentence and second paragraph in Conclusions are not suitable; suggest deleting them.
7) Please include the SD or SEM in Tables 3,4,5.
8) The abbreviations should be defined when it first appears. Such as TAC, and SAA. And these abbreviations should be defined in the Tables.
Author Response
Response to Reviewer 3 Comments
Dear Reviewer,
Thank you very much for reviewing our manuscript Effect of plant bioactive compounds supplemented in transition dairy cows on the metabolic and inflammatory status. Please, find below our responses to your comments and suggestions. The manuscript has been edited accordingly.
Sincerely,
Dana Kumprechtová, on behalf of the team of authors
Point 1: Please list out the detailed animal information such as DIM, milk yield, and parity in average +-SD. You have 22 primiparous and 24 multiparous cows, are they allocated to the two groups on average?
Response 1: The information has been added as Table 6 (L251).
Point 2: Was the feed intake measured?
Response 2: Feed intake was not measured (L265). It was impossible to measure feed intake under field conditions.
Point 3: Why do you select 25 g/cow/day? Did you conduct a pre-trial or refer to a previous study to use this dose?
Response 3: Explanation is provided on L246, 247 (the company´s recommendations and previous studies using some PhytoAx´cell components (Gobert, M.; Martin, B.; Ferlay, A.; Chilliard, Y.; Graulet, B.; Pradel, P.; Bauchart, D.; Durand, D. Plant Polyphenols Associated with Vitamin E Can Reduce Plasma Lipoperoxidation in Dairy Cows given N-3 Polyunsaturated Fatty Acids. J. Dairy Sci. 2009, 92, 6095–6104, doi:10.3168/jds.2009-2087)
Point 4: Suggest changing Table 6 to percentage of dry matter basis, not the kg in the fresh matter.
Response 4: Unfortunately, it was not feasible to regularly measure dry matter contents of different dietary components, neither total diet´s DM. The Table 7 (Table 6 was changed to Table 7 because the randomization Table 6 was added) heading was changed to “Composition of the control diet on a fresh matter basis”
Point 5: The total polyphenols in Phyto Ax’Cell is 20.9g/kg, so what are the other components?
Response 5: Crude constituents were added (L288, 289).
Point 6: The first sentence and second paragraph in Conclusions are not suitable; suggest deleting them.
Response 6: Conclusion was changed.
Point 7: Please include the SD or SEM in Tables 3,4,5.
Response 7: SEM has been added in Tables 3,4,5.
Point 8: The abbreviations should be defined when it first appears. Such as TAC, and SAA. And these abbreviations should be defined in the Tables.
Response 8: The text has been edited accordingl

Round 2
Reviewer 1 Report
Line no. Avoid multiple citations for one sentence. If not necessary, remove references from the text Eg. 33, 37
Suggested to remove the terms – We, I from the manuscript text, Eg. We observed… Instead of using it as “ The study observed that…”
Author Response
Dear Reviewer,
Thank you very much for reviewing our manuscript Effect of plant bioactive compounds supplemented in transition dairy cows on the metabolic and inflammatory status. Please, find below our responses to your comments and suggestions. The manuscript has been edited accordingly.
Sincerely,
Dana Kumprechtová, on behalf of the team of authors
Point 1: Avoid multiple citations for one sentence. If not necessary, remove references from the text Eg. 33, 37
Response 1: We have removed multiple citations per sentence.
Point 2: Suggested to remove the terms – We, I from the manuscript text, Eg. We observed… Instead of using it as “ The study observed that…”
Response 2: We have removed “we” from the manuscript: L58, L141, L161, L172

Reviewer 3 Report
Accept
Author Response
Dear Reviewer,
Thank you very much for accepting the revised manuscript.
Kind regards,
Dana Kumprechtová